# Passive and active DNA methylation and the interplay with genetic variation in gene regulation

**Maria Gutierrez-Arcelus**[1,2,3], **Tuuli Lappalainen**[1,2,3], **Stephen B Montgomery**[1,4], **Alfonso Buil**[1,2,3], **Halit Ongen**[1,2,3], **Alisa Yurovsky**[1,2,3], **Julien Bryois**[1,2,3], **Thomas Giger**[1,2], **Luciana Romano**[1,2], **Alexandra Planchon**[1,2,3], **Emilie Falconnet**[1,2], **Deborah Bielser**[1,2], **Maryline Gagnebin**[1,2], **Ismael Padioleau**[1,2,3], **Christelle Borel**[1,2], **Audrey Letourneau**[1,2], **Periklis Makrythanasis**[1,2], **Michel Guipponi**[1,2], **Corinne Gehrig**[1,2], **Stylianos E Antonarakis**[1,2]*, **Emmanouil T Dermitzakis**[1,2,3]*

[1]Department of Genetic Medicine and Development, University of Geneva Medical School, Geneva, Switzerland; [2]Institute of Genetics and Genomics in Geneva, Geneva, Switzerland; [3]Swiss Institute of Bioinformatics, Geneva, Switzerland; [4]Departments of Pathology and Genetics, Stanford University, Stanford, United States

**Abstract** DNA methylation is an essential epigenetic mark whose role in gene regulation and its dependency on genomic sequence and environment are not fully understood. In this study we provide novel insights into the mechanistic relationships between genetic variation, DNA methylation and transcriptome sequencing data in three different cell-types of the GenCord human population cohort. We find that the association between DNA methylation and gene expression variation among individuals are likely due to different mechanisms from those establishing methylation-expression patterns during differentiation. Furthermore, cell-type differential DNA methylation may delineate a platform in which local inter-individual changes may respond to or act in gene regulation. We show that unlike genetic regulatory variation, DNA methylation alone does not significantly drive allele specific expression. Finally, inferred mechanistic relationships using genetic variation as well as correlations with TF abundance reveal both a passive and active role of DNA methylation to regulatory interactions influencing gene expression.

*For correspondence: stylianos. antonarakis@unige.ch (SEA); emmanouil.dermitzakis@unige.ch (ETD)

**Reviewing editor**: Chris P Ponting, University of Oxford, United Kingdom

## Introduction

DNA methylation is an essential (*Li et al., 1992*) epigenetic mark whose role in gene regulation and its dependency on genomic sequence and environment is not yet fully understood (*Jones, 2012*; *Schubeler, 2012*). DNA methylation in vertebrates occurs most commonly in cytosines that are adjacent to guanines (CpG sites). Mammalian DNA methylation levels are generally high, although CpG rich regions, called CpG islands (CGI), appear mostly unmethylated (*Bird, 2002*; *Weber et al., 2007*; *Lister et al., 2009*). The mechanisms of de novo methylation and maintenance of methylation patterns are well known (*Shoemaker et al., 2011*), and alterations of these can cause severe diseases (*Robertson, 2005*). Even though it was originally reported to be involved in gene silencing (*Holliday and Pugh, 1975*; *Riggs, 1975*), more recent studies have found that DNA methylation can also be positively correlated to gene transcription when found in gene bodies (*Hellman and Chess, 2007*; *Lister et al., 2009*). Additionally, its participation in gene expression is proving to be highly variable, ranging from marking alternative intra-genic promoters (*Maunakea et al., 2010*), to being affected by

**eLife digest** Variations occur throughout our genome. These variations can cause genes to be expressed (switched on) in slightly different ways among individuals. Moreover, the same gene can also be expressed in different ways in different cells within an individual. A third level of variation is supplied by epigenetic markers: these are molecules that bind to the DNA at specific points and can have profound effects on the expression of nearby genes. One such epigenetic marker is the addition of a methyl group to a cytosine base, a process that is known as DNA methylation.

DNA methylation usually happens when a cytosine base is next to a guanine base, forming a CpG site. In mammals, most CpG sites have methyl groups attached, although regions with a lot of CpG sites (called CpG islands) are mostly unmethylated. Initial studies suggested that methylation prevented particular genes from being expressed, but more recent work has indicated that methylation can be associated with both reduced and increased expression of genes. Moreover, it is not clear if this association is active (i.e., changes in methylation drive changes in gene expression) or passive (DNA methylation is the result of gene regulation).

Now, Gutierrez-Arcelus et al. have carried out a large-scale study to clarify the relationships between three different types of gene-related variations among individuals. They extracted fibroblasts, T-cells and lymphoblastoid cells from the umbilical cords of 204 babies, and analysed them for variations in DNA sequence, gene expression and DNA methylation. Their results show that the associations between the three are more complex than was previously thought.

Gutierrez-Arcelus et al. show that the mechanisms that control the association between the variations in DNA methylation and gene expression in individuals are likely to be different to those that are responsible for the establishment of methylation patterns during the process of cell differentiation. They also find that the association between DNA methylation and gene expression can be either active or passive, and can depend on the context in which they occur in our genome. Finally, where the two copies or alleles of a gene are not equally expressed in a given cell, the difference in expression is primarily regulated by DNA sequence variation, with DNA methylation having little or no role on its own. Equally complex interactions and effects are expected in further studies of genetic and epigenetic variation.

transcription factors (TFs) at enhancers (*Stadler et al., 2011*) or it-self affecting the binding of TFs such as MYC (*Prendergast and Ziff, 1991*). Hence, whether DNA methylation is a consequence of gene regulation, or whether it controls gene expression changes—that is, whether it plays a passive or an active role in gene regulation—still remains a topic of debate (*Schubeler, 2012*).

Additionally, DNA methylation can be affected by environment (*Kaminsky et al., 2009*) but it has also been proven that regions in the genome can autonomously determine DNA methylation states (*Lienert et al., 2010*). Furthermore, studies looking at natural DNA methylation variation in human populations have shown that genetic variation influences DNA methylation levels in different cell-types (*Gibbs et al., 2010*; *Zhang et al., 2010*; *Bell et al., 2011*), but the mechanisms by which this occurs are far from clear. In these and other studies (*Kulis et al., 2012*), the association between DNA methylation and gene expression in a population context, where the same gene and same methylation site can be compared across multiple individuals, has been reported to be both positive and negative. Overall, the nature of the relationships among genetic variants, DNA methylation and gene expression are still unclear despite some initial efforts (*van Eijk et al., 2012*).

In this study we dissect the mechanistic relationships between inter-individual DNA methylation and gene expression variation using DNA sequence variability and TF abundance measured by RNA-Seq. By assaying in a high resolution and genome wide level these three layers of information in three different cell-types originating from the same set of individuals, we are able to study the role of DNA methylation variation in different dimensions. Our results reveal a picture where DNA methylation variable sites are mechanistically associated to gene expression in complex and context dependent ways that can be of passive or active nature. We further highlight some of the mechanisms by which passive DNA methylation may occur and how this role can interplay with genetic variation.

# Results

## Associations of genetic variation, DNA methylation and gene expression

We use the GenCord collection (*Dimas et al., 2009*) of umbilical cord samples from 204 newborn babies of central European descent, from which we derived three cell-types: fibroblasts (primary cells), T-cells (primary cells) and lymphoblastoid cells (immortalized cell lines, LCLs) (*Figure 1*). We genotyped each individual for 2.5 million SNPs, and sequenced the poly-A transcriptome of all three cell-types from the 204 individuals yielding a median of 16 million exonic reads per sample. We subsequently removed samples from 18 to 21 genetic or expression outliers, yielding a final set of 183–186 individuals per cell type (*Figure 1—figure supplements 1 and 2*). The assayed SNPs were imputed to the Phase 1 release of the 1000 genomes project (*Abecasis et al., 2012*) yielding a set of 6.9 million SNPs. We obtained normalized expression levels for 70,800–76,870 exons belonging to 12,265–12,863 genes (*Figure 1—figure supplements 3 and 4*). DNA methylation levels were measured using bisulfite-conversion and hybridization to a bead chip, assaying 416,118 CpG sites in 66–118 samples. Normalized methylation levels of CpG sites range from 0 to 1, reflecting the percentage of methylation per site (β-value; *Figure 1—figure supplements 5–7*). In total, we analyzed 66–186 samples per cell-type and assay, belonging to 195 individuals (*Table 1*; see 'Materials and methods').

At 10% FDR, we identify 2115–3372 expression quantitative trait loci (eQTLs) using a 1 Mb window to either side of the TSS. We discover 14,189–32,318 methylation QTLs (mQTLs) using a 5-kb window to either side of the CpG site. We find 1541–17,267 significant expression to methylation associations (eQTMs) using a 50-kb window around the TSS; these pertain to 596–3838 genes and 970–6846 CpG sites (*Table 1*, *Figure 1*).

## Orthogonal roles of developmental and inter-individual DNA methylation variation

DNA methylation at promoter regions is widely known to correlate negatively with gene expression levels when looking at comparisons across genes (*Jones, 2012*). We observe the same pattern in our study looking at the promoter regions of all genes of each individual separately and for each cell-type (*Figure 2—figure supplement 1*). The across individual methylation-gene expression associations (eQTMs) however appear to be either positive or negative, even for DNA methylation sites in promoter regions. Hence, we hypothesized that methylation sites in promoter regions from positive (pos) and negative (neg) eQTMs contribute differently (positively and negatively, respectively) to gene expression levels across genes. Contrary to our hypothesis, we find that methylation sites correlate negatively with gene expression across genes independently of whether they correlate positively or negatively with gene expression across individuals (*Figure 2A*; Spearman correlation coefficient, $rho = -0.11$, $p=1.1 \times 10^{-4}$, and $rho = -0.10$, $p=1.7 \times 10^{-13}$, respectively; see also *Figure 2—figure supplement 2*). The strength of these negative correlations, despite involving a subset of genes, is comparable to the one found at a genome-wide level, correlating all expressed genes with their promoter DNA methylation status per individual (*Figure 2—figure supplement 1C*). These results suggest that the mechanisms and processes underlying inter-individual DNA methylation variation associated to gene expression are at least partly independent of the mechanisms involved in the establishment of the repressive mark of promoter DNA methylation across genes during development and differentiation.

Differentially methylated regions among cell-types have been shown to play important roles in cell differentiation and tissue-specific regulation (*Meissner et al., 2008*; *Schmidl et al., 2009*; *Hodges et al., 2011*; *Li et al., 2012*). Hence, we asked whether differentially methylated sites across the three cell-types are more likely to be functionally relevant when variable within a population, hence more often associated to gene expression (eQTMs) or genetic variation (mQTLs) than non-differentiated sites across cell-types. We measured the level of differentiation for each site by calculating its median methylation level in each cell-type and then calculating the coefficient of variation between those three medians, which is a measure of variability that controls for the mean level of methylation (*Figure 2—figure supplement 3A*). We find that as differentiation per methylation site increases, a higher proportion of eQTMs and mQTLs are observed (*Figure 2B*). This is statistically supported by the fact that the tissue differentiation level of methylation sites involved in eQTMs and mQTLs is significantly higher than for non-eQTM and non-mQTL sites, respectively (all p values$<2.2 \times 10^{-16}$, Wilcoxon test, *Figure 2—figure supplement 3B,C*). These results show that the same methylation sites marking or contributing

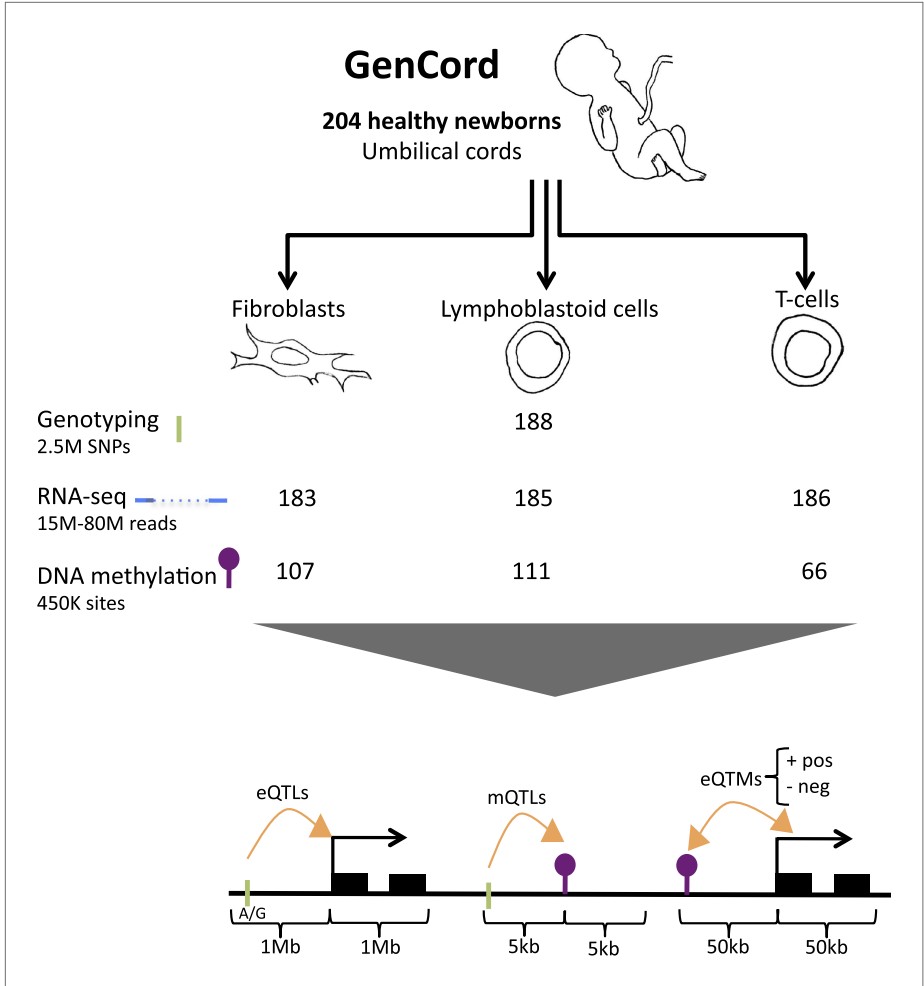

**Figure 1**. GenCord project scheme. We collected umbilical cord and cord blood samples from 204 newborn babies, from which we derived three cell-types: fibroblasts, lymphoblastoid cells and T-cells. Genotyping, RNA-sequencing and DNA methylation levels were assayed. The number of samples without genetic and technical outliers is indicated for each assay and each cell-type. We then correlated and utilized different properties of all datasets in order to assess: expression Quantitative Trait Loci (eQTLs), methylation QTLs (mQTLs), positive (pos) and negative (neg) expression Quantitative Trait Methylation (eQTMs). Green ticks represent Single Nucleotide Polymorphisms (SNPs), purple lollipops represent methylation sites, black boxes represent exons and orange arrows depict associations between two data-types. Shown are the maximum distances between each pair of variables tested. See *Figure 1—figure supplement 1–7* for data processing and quality checks.

The following figure supplements are available for figure 1:

**Figure supplement 1**. Genetic outliers removed from analyses involving genetic variation.

**Figure supplement 2**. Number of reads per sample, before removing technical outliers.

**Figure supplement 3**. Covariates for which expression data was corrected.

**Figure supplement 4**. Pair wise correlations among individuals before and after covariate correction.

**Figure supplement 5**. Normalized β-value and variance across individuals.

**Figure supplement 6**. Normalized β-value pair wise correlations between individuals.

**Figure supplement 7**. β-value distributions in distinct genomic features for expressed and non-expressed genes.

**Table 1.** Summary of main association analyses in GenCord. See *Table 1–source data 1*

| | Test | Samples | Window size | Unit | FDR (%) | Nominal *p* value | Fibroblasts | LCLs | T-cells |
|---|---|---|---|---|---|---|---|---|---|
| eQTLs | Genotypes and expression | 183 (F); 185 (L); 186 (T) | 1 Mb | Genes | 10 | $2.2 \times 10^{-5}$ (F); $3.2 \times 10^{-5}$ (L); $1.8 \times 10^{-5}$ (T) | 2433 | 3372 | 2115 |
| mQTLs | Genotypes and methylation | 107 (F); 111 (L); 66 (T) | 5 kb | Methylation sites | 10 | $4.4 \times 10^{-4}$ (F); $7.9 \times 10^{-4}$ (L); $1.3 \times 10^{-3}$ (T) | 14,189 | 22,411 | 32,318 |
| eQTMs | Methylation and expression | 110 (F); 118 (L); 66 (T) | 50 kb | Genes | 10 | $7.6 \times 10^{-5}$ (F); $7 \times 10^{-4}$ (L); $6.9 \times 10^{-4}$ (T) | 596 | 3680 | 3838 |

The following source data are available for table 1:
**Source data 1**. Significant eQTL, mQTL and eQTM associations found in each cell-type.

to tissue differentiation are participating in inter-individual variability that is highly determined by genetic variation and associated to gene expression. This could indicate that the establishment of differentially methylated regions during development could be delineating a backbone where local inter-individual changes may occur.

## Context-specific DNA methylation

In order to further understand the nature of these inter-individual changes we analyzed their participation in different genomic contexts. We find a significant increased presence of negative compared to positive eQTMs in CTCF binding sites, enhancers and promoters, with this increase being higher in non-CpG island (CGI) promoters compared to CGI promoters (*Figure 2C*). Interestingly, we discover a significant depletion of mQTLs in CGI promoters, and a significant enrichment of mQTLs in non-CGI promoters (*Figure 2D*). This shows that methylation sites in non-CGI promoters are under a stronger genetic control than at CGI promoters, where methylation is in general robustly maintained at low levels. Nevertheless, genetic variation in CpG islands commonly affects gene expression, since we find that eQTLs are significantly enriched at these genomic regions (all p values<0.03). Overall, these results suggest that the role of DNA methylation can be highly dependent on the genomic and functional context.

## DNA methylation and allele specific expression

Allele specific expression (ASE), seen as a signal of regulatory difference between two haplotypes of an individual, can in theory be driven either by genetic regulatory variation or epigenetic (in)activation of one of the two alleles, for example, by DNA methylation. In order to test whether ASE is caused by genetic variation or differential DNA methylation, we compared the magnitude of allelic imbalance in eQTL genes between individuals who are heterozygous for the eQTL and their respective homozygotes. Similar to other studies (*Dimas et al., 2009*; *Pickrell et al., 2010*; *Montgomery et al., 2010*), at a genome-wide level a significantly greater allelic imbalance is associated with heterozygote eQTLs in all three cell-types (p values<$3.0 \times 10^{-8}$, *Figure 3A*). In order to test whether ASE is driven by haplotype differences in DNA methylation alone, potentially represented by semimethylated sites (partially methylated, β-value >0.3 and <0.7), we analyzed the genes having eQTM methylation sites that are not associated with SNPs (filtered out SNP-methylation correlations with nominal p<0.01). A comparison of allelic imbalance between individuals with semimethylated eQTMs, and homomethylated (i.e., fully methylated, β-value > 0.7, or unmethylated, β-value < 0.3) eQTMs, revealed no significant differences (p values>0.42, *Figure 3A*). Additionally, ASE driven by heterozygote eQTLs is significantly higher than that of semimethylated eQTMs (p values<$7.5 \times 10^{-3}$, *Figure 3A*). Furthermore, as a positive control, we observe that allelic imbalance in imprinted genes, known to be driven by allelic DNA methylation, is significantly higher than in homomethylated eQTMs (p values<$9.7 \times 10^{-4}$, *Figure 3A*). Thus, while ASE is shown to be significantly driven by genetic variation (or by DNA methylation in imprinted genes), we find no evidence in our data of methylation alone contributing to ASE. This argues that DNA methylation is rarely allele-specific in the absence of DNA sequence variation effects, and while we cannot exclude other possible epigenetic sources of ASE, the results suggest that the widespread ASE across the genome may be primarily driven by common and rare genetic regulatory variants.

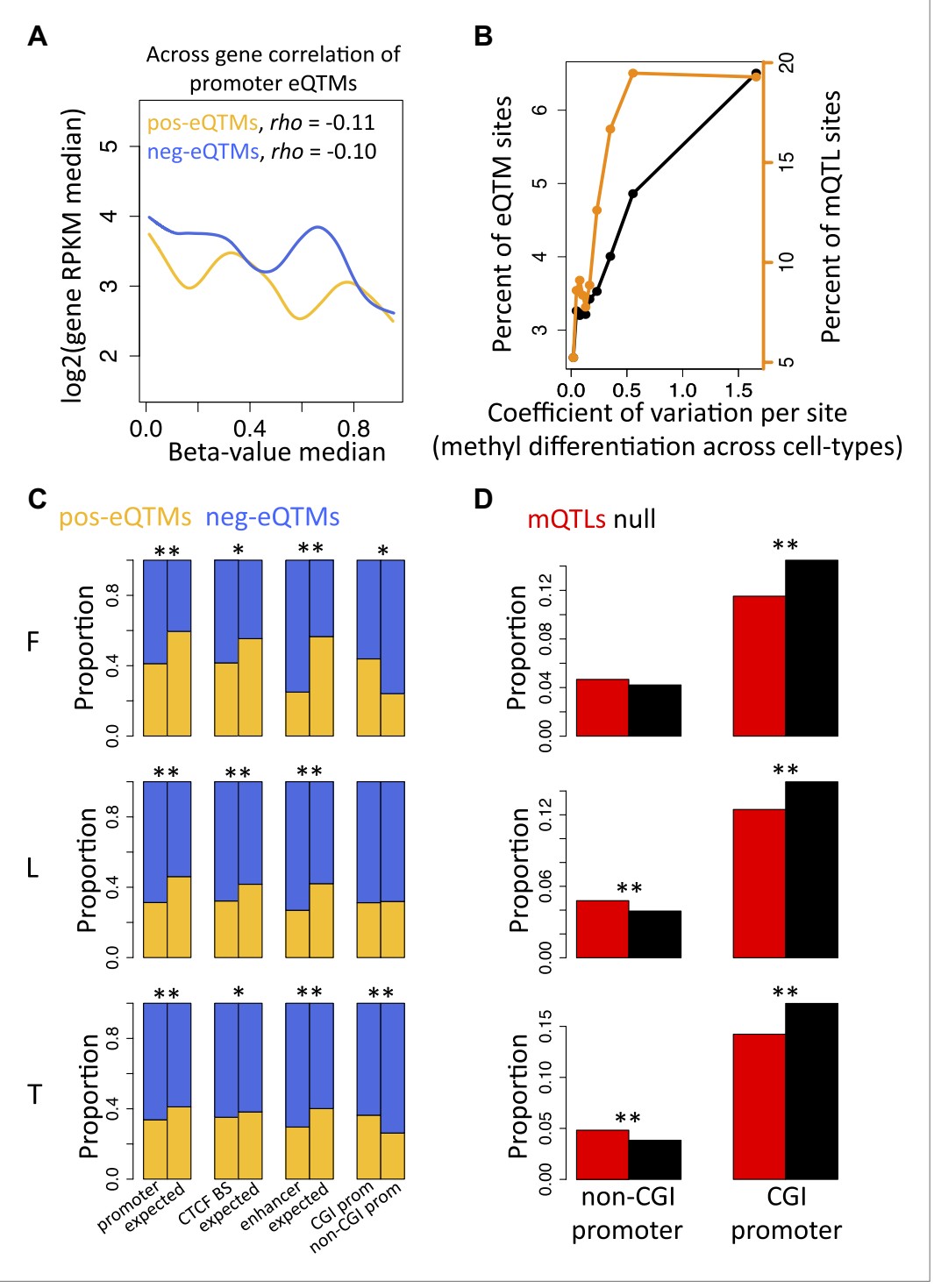

**Figure 2**. Inter-individual DNA methylation variation in cell-type differentiation and in different genomic contexts. (**A**) The median methylation level of promoter eQTM sites (x-axis) correlates negatively with across gene median number of reads per kilobase per million reads (RPKM) irrespective of whether they are pos-eQTMs (yellow, N = 1149) or neg-eQTMs (blue; N = 5112). Spearman correlation coefficient *rho* is indicated in the plot with p=1.1 × 10^{-4} and p=1.7 × 10^{-13} for pos and neg-eQTMs, respectively. See ***Figure 2—figure supplements 1 and 2***. (**B**) As the level of cell-type methylation differentiation increases (x-axis), a larger proportion of sites are associated to gene expression (eQTMs, left y-axis) and affected by genetic variation (mQTLs, right y-axis). Proportions are plotted by 10 bins each containing 10% of the data (0.1 quantiles). Level of methylation differentiation is measured for each site as the
*Figure 2. Continued on next page*

*Figure 2. Continued*

coefficient of variation of the median methylation level per cell-type. See *Figure 2—figure supplement 3*. (**C**) Proportion of eQTMs that are positive (pos-eQTMs, yellow) or negative (neg-eQTMs, blue) overlapping vs non-overlapping (expected) distinct genomic features (promoters, CTCF binding sites, enhancers), or overlapping CpG island promoters (CGI prom) vs overlapping non-CpG island promoters (non-CGI prom). For T-cells there are no CTCF or chromatin ChIP-seq data available so the data of an LCL were used instead (see Materials and methods). (**D**) For each non-CGI and CGI promoters (x-axis), the proportion (y-axis) of overlapping mQTLs was calculated (red bars) and was compared to the proportion of overlapping null SNPs (black bars). One star indicates p<0.05, two stars indicate $p < 1 \times 10^{-6}$, Fisher's exact test.

The following figure supplements are available for figure 2:

**Figure supplement 1**. Correlation between promoter DNA methylation and across gene expression.

**Figure supplement 2**. pos and neg-eQTMs correlation with across gene expression.

**Figure supplement 3**. Tissue-specific methylation is enriched with eQTMs and mQTLs.

## Synergistic interactions between DNA methylation and genetic variation on gene expression

We then sought to explore the mechanistic relationships among DNA methylation, gene expression and genetic variation. We first tested whether there was a significant enrichment of synergistic interactions between genetic variants and DNA methylation on gene expression using linear regression. We selected for each exon all eQTLs with $p < 1 \times 10^{-4}$ that fell in independent recombination intervals and all eQTMs with p<0.001. To avoid artificial inflation of significant interactions, we filtered out any exon-SNP-meth triplets where the SNP and methylation site correlated with p<0.05. To avoid spurious interactions caused by outliers, we further filtered out cases where there were less than four individuals homozygous for the minor allele of a SNP. Finally, to further account for remaining correlation between the SNP and the methylation we permuted the expression values 1000 times to infer the 95% confidence intervals and assess the significance of the enrichment of low p values. Synergistic interactions are enriched in LCLs and T-cells (*Figure 3B*) with $\pi_1$ being 4.3% and 9.3% (all p<0.001), which reflects the percent of estimated true positives from the p value distributions (*Storey and Tibshirani, 2003*). In fibroblasts, although the p value distribution is not above the 95th confidence limit (*Figure 3B*), the observed $\pi_1$ is 12.4% (p<0.001), 7.2 times larger than the top permuted $\pi_1$, which suggests a significant enrichment of interactions. Finally, we find 3, 91 and 14 individually significant interactions in fibroblasts, LCLs and T-cells, respectively, at 10% false discovery rate (FDR). Overall, these results reflect the interdependency of genetic and epigenetic variation to determine gene expression levels.

## Passive and active roles of DNA methylation in gene regulation

We further dissected the causative relationships between DNA methylation and gene expression by considering that the SNP triggers the causal network since its state is not modifiable in time. We used Bayesian Network (BN) construction and relative likelihood (see 'Materials and methods') to test which of the three possible causative models depicted in *Figure 4A* and described below is the most likely given the data for each set of variables. Under the 'INDEP' (independent) model, a SNP affects independently gene expression and DNA methylation (passive role of methylation). Under the 'SME' (SNP-methylation-expression) model, the SNP affects methylation, which then affects gene expression (active role). The 'SEM' (SNP-expression-methylation) model requires that a SNP affects gene expression, and expression then affects methylation (passive role). We tested the relative likelihood of these models choosing a non-biased approach where at least two of the three pairwise correlations between genetic variation, gene expression and DNA methylation are significant, yielding 831–2928 SNP-methyl-exon triplets tested per cell-type. All three models occur, depending on cell-type and sign of correlation between DNA methylation and gene expression (*Figure 4B*, *Figure 4—figure supplement 1A*). The INDEP model, that reflects a passive role for DNA methylation, is the most likely model in fibroblasts and LCLs. However, in T-cells the SME model, where methylation takes an active causative role, is the major contributor. Note that in the SEM model, where methylation is passive being influenced by gene expression levels, a higher likelihood of pos-eQTMs is found compared to the SME

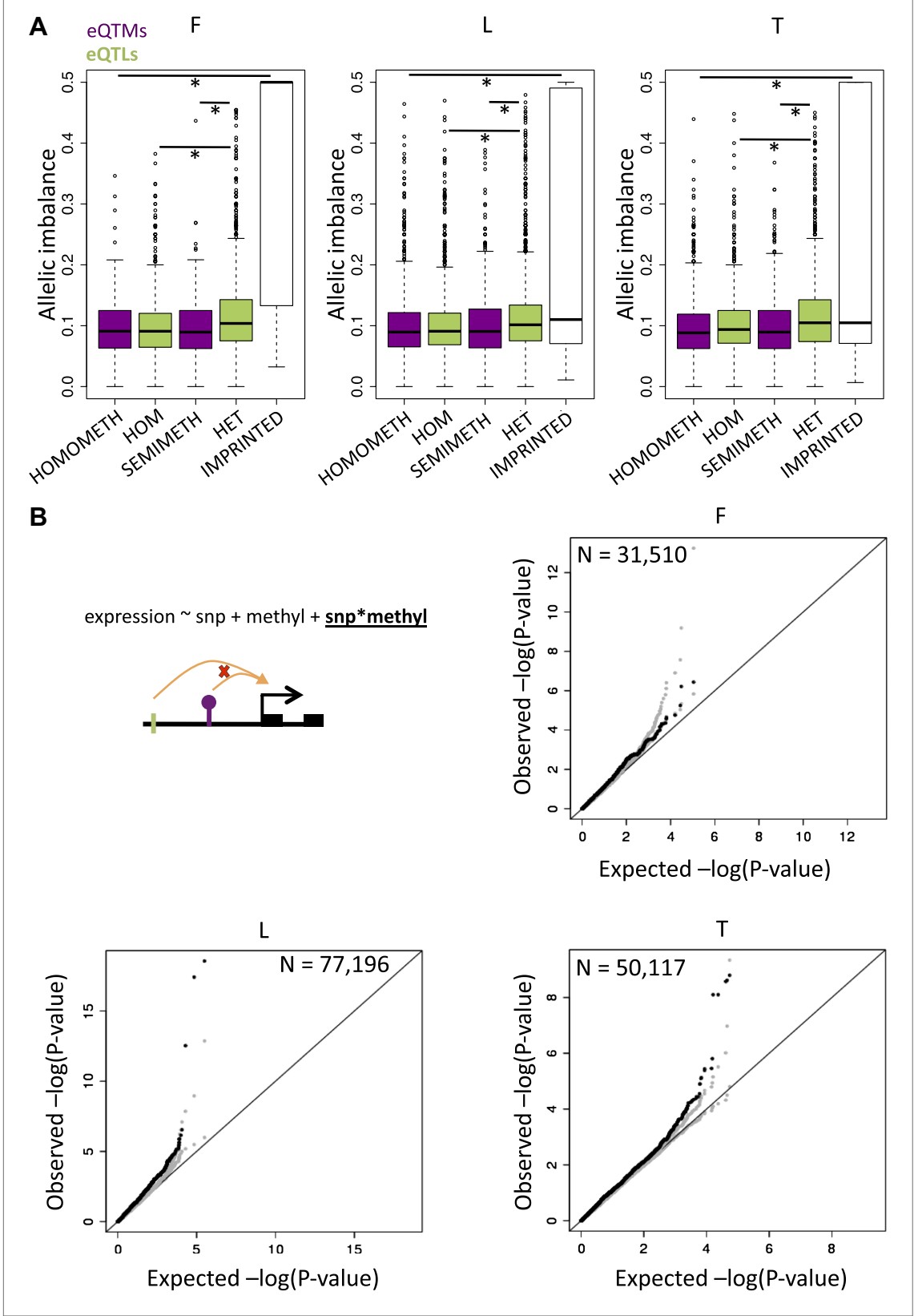

**Figure 3**. DNA methylation associated to gene expression is not significantly allelic and can interact with genetic variation. (**A**) In green are depicted the distributions of allelic imbalance (i.e., absolute distance from the expected 0.5 ratio) of assayable heterozygote sites in eQTL genes of individuals that are homozygous (HOM) or heterozygous (HET) for the eQTL. The difference between distributions is significant in all cell-types with p<2.2 × 10⁻¹⁶, p=2.7 × 10⁻¹⁵

*Figure 3. Continued on next page*

*Figure 3. Continued*

and p=3.0 × 10⁻⁸ in fibroblasts (F), LCLs (L) and T-cells (T), respectively (Wilcoxon test). This strongly indicates that allele specific expression is significantly driven by genetic variation. In purple are shown the distributions of allelic imbalance of assayable heterozygote sites in eQTM genes (excluding methylation sites affected by genetic variation) of individuals that are homomethylated (HOMOMETH, i.e., fully methylated, β-value > 0.7, or unmethylated β-value < 0.3) or semimethylated (SEMIMETH, i.e., β-value >0.3 and <0.7) for the eQTM site. The difference between distributions is not significant in any cell-type, with p=0.79, p=0.42 and p=0.49 in F, L, T, respectively. The difference between distributions of HET eQTLs and SEMIMETH eQTMs is significant in all cell-types with p=7.5 × 10⁻³, p=3.4 × 10⁻⁷, p=1.1 × 10⁻¹³, in F, L, T, respectively. The difference between distributions of IMPRINTED genes and HOMOMETH eQTMs is significant in all cell-types with p=6.9 × 10⁻³⁴, p=9.7 × 10⁻⁴, p=8.5 × 10⁻⁵ in F, L, and T, respectively. This shows that allele specific expression is not significantly driven by DNA methylation that is not affected by genetic variation. (**B**) Using linear regression we tested the interaction term as shown in the illustrated formula for the effects of SNPs (green tick) and methylation sites (purple lollipop) on gene expression (black squared arrow and boxes). Qqplots illustrate the enrichment of low synergistic interaction observed p values (black), together with the 5th and 95th confidence limits based on expression permutations (gray) with respect to the expected uniform distribution.

model. Overall, these results suggest that DNA methylation can be both active, by being a likely cause of gene expression variation levels, or passive, by being a consequence or an independent mark of gene expression levels.

In order to obtain a set of high confidence calls for each of the models, we have required that the BN calls are confirmed by an independent method called the Causal Inference Test (*Millstein et al., 2009*) (see details in 'Materials and methods'). From the total number of tests, 61%, 36% and 27% were called as high confidence (HC) in fibroblasts, LCLs and T-cells, respectively (*Figure 4—source data 1*). The relative frequencies of these HC calls look similar to the general relative likelihood space of the models (*Figure 4—figure supplement 1B*). As an example of a HC INDEP model, we identified in fibroblasts a case occurring at the promoter of gene *DPYSL4* and involving a methylation site associated to age and age rate in blood samples (*Hannum et al., 2013*). In this example, SNP rs12772795 affects independently the DNA methylation status of site cg05652533 and the expression level of gene *DPYSL4*, possibly via an effect on the binding of CTCF nearby (given binding peak reported), which is a factor known to alter DNA methylation levels locally (*Stadler et al., 2011*). As an example of an SME model, we have identified in T-cells SNP rs1362125 that could be affecting the binding of SP1 (overlapping peak reported), which is a factor shown to confer methylation protection (*Boumber et al., 2008*). This could then alter the methylation state of site cg24703717 (78bp away) that falls in a YY1 binding peak, a factor whose binding is known to be sensitive to DNA methylation levels (*Kim et al., 2003*). Hence methylation could actively alter binding of YY1, negatively affecting the expression of the gene *HLA-F*. Finally, we would like to highlight in LCLs an SEM scenario in which SNP rs3733346, located in a DNAse hypersensitive site, affects the expression of gene *DGKQ*, whose transcriptional activity could then be influencing positively DNA methylation levels of site cg00846425 located in its gene body, as has been suggested to be a possible phenomenon for gene body DNA methylation (*Hahn et al., 2011*).

## DNA methylation associated to transcription factor abundance

To understand potential molecular causes for the passive role of methylation in the INDEP model we postulated that SNPs could be influencing binding levels of DNA-binding factors (hereon called transcription factors or TFs), which in turn affect the methylation status of a site in parallel to their effect on gene expression. It has been shown in mice (*Stadler et al., 2011*) that TFs can influence DNA methylation levels near their binding sites (TFBSs). In addition, by correlating expression levels of TFs with DNA methylation at their TFBSs in different cell-types, this pattern has also been observed for some TFs in human tissues (*Thurman et al., 2012*). In order to test whether human inter-individual differences of TF expression levels would affect DNA methylation that is itself associated to gene expression we used the ENCODE dataset of TFBSs based on ChIP-seq assays for 111 TFs (*Bernstein et al., 2012*; *Gerstein et al., 2012*), and correlated their expression levels with eQTM methylation levels found at their reported binding sites. We found a strong enrichment of significant TF-methylation associations, with π₁ being 18%, 9% and 25%, in fibroblasts, LCL, and T-cells, respectively (*Figure 4—figure supplement 2*). At 10% FDR, we find significant associations for 27, 47 and 99 different TFs in fibroblasts, LCL and T-cells, respectively. A strong enrichment of significant associations can also be appreciated for each TF individually as depicted in *Figure 4C*. We further tested whether genetic

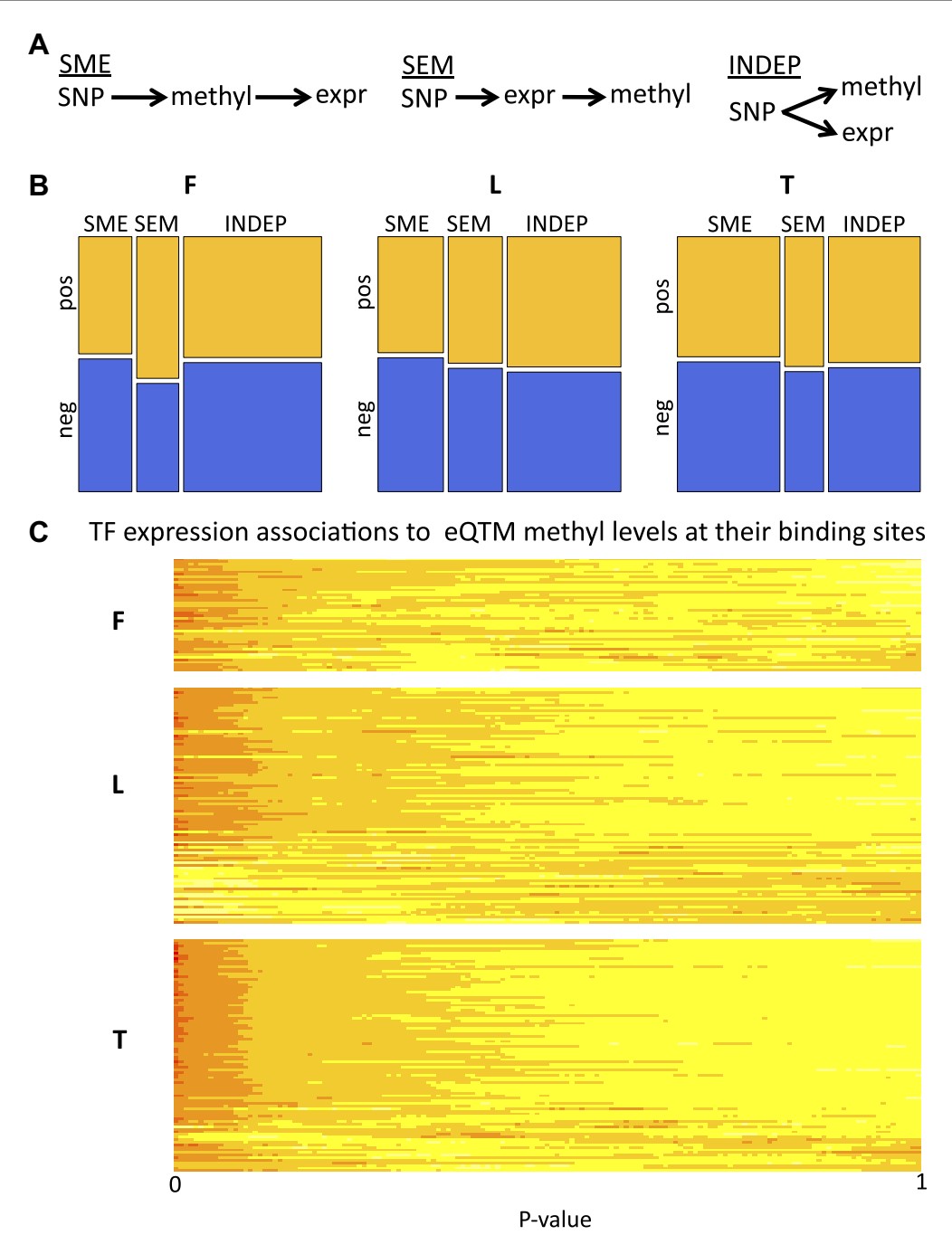

**Figure 4**. Passive and active roles of DNA methylation in gene regulation. (**A**) Illustration of the three possible causative models tested of mechanistic relationships between genetic variation (SNP), DNA methylation (methyl) and gene expression (expr). Arrows indicate the causal direction of effects. The name of each model is underlined. (**B**) Mosaicplots illustrate the relative likelihoods of each model (x-axis), partitioned by the relative likelihoods of those involving pos-eQTMs (yellow) and neg-eQTMs (blue; y-axis), in fibroblasts (F), LCLs (L) and T-cells (T). The three types of models (INDEP, SME and SEM) are present in the three cell-types, suggesting that DNA methylation can have both active and passive roles in gene regulation. See *Figure 4 –figure supplement 1* and *Figure 4–Source data 1*. (**C**) Heatmap of p value relative frequency distributions of spearman correlations between transcription factors (TF) and DNA methylation levels of eQTMs at their binding sites, sorted by $\pi_1$. The enrichment of significant associations can be appreciated by the accumulation of reddish colors, reflecting higher relative frequencies, at low p values, and yellowish colors, reflecting lower relative frequencies, at higher p values. These
*Figure 4. Continued on next page*

*Figure 4. Continued*

results highlight one of the possible mechanisms of a passive role of DNA methylation regarding gene expression. See *Figure 4—figure supplements 2 and 3*.

The following source data and figure supplements are available for figure 4:

**Source data 1**. High confidence calls for INDEP, SME and SEM models in each cell-type.

**Figure supplement 1**. Passive and active roles of DNA methylation on gene regulation.

**Figure supplement 2**. Associations between TF abundance and DNA methylation at their target binding sites.

**Figure supplement 3**. Interactions between SNPs and transcription factor levels on DNA methylation.

---

variants that could potentially alter TF binding would interact with TF abundance on their effect on DNA methylation levels (see 'Materials and methods'). Despite the limited number of cases that could be tested (N = 114), we find an enrichment of low p values for such interactions, with $\pi_1$ estimated at 15% of true positives, and the top interaction involving the TF c-Jun in T-cells (p=1 × 10$^{-4}$, *Figure 4—figure supplement 3*). These results suggest that TF levels could be influencing DNA methylation and gene expression levels simultaneously. It is possible that in some cases this could occur via the interaction with DNA sequence variation. This could illustrate a realistic model for passive but at the same time correlated levels of DNA methylation with gene expression.

## Discussion

In this study, we have shown that inter-individual DNA methylation changes are mechanistically associated to genetic variation and gene expression in complex and context dependent ways that can be of passive or active nature. DNA methylation levels can depend on TF binding (*Stadler et al., 2011*). TF binding levels in turn can be determined by both inter-individual differences of TF abundance and genetic variants at their binding sites. This picture shows how DNA methylation can be linked to genetic variation, and can participate in gene regulation in a passive manner. This scenario is quite different from the often-assumed simple model of genetic variation affecting the epigenome, which in turn affects the transcriptome, and furthermore these relationships can be of different nature depending on tissue and genomic region. This is likely to be true for other epigenetic mechanisms as well, and properly characterizing these relationships will enhance our understanding of genome function and complex disease.

Additionally, our results suggest that non-genetically determined DNA methylation is rarely allelic but rather defined at the whole cell level. That is, when we observe a population of cells to be semi-methylated the most likely scenario is that only a fraction of the cells are fully methylated rather than one of the alleles in each of the cells is methylated. Nevertheless, given the limitations of our data, further studies addressing this question are needed to confirm our results.

Further, we show that methylation sites that represent a repressive state with respect to tissue differentiation can also participate in positive and negative associations with gene expression when looking at inter-individual variability. DNA methylation as a whole emerges both as marker and determinant of cellular identity and may provide a platform by which genetic variation and other variable factors among individuals (e.g., TFs) exert their effect to differentially influence cellular processes.

## Materials and methods

### Sample collection and cell growth

Fifty-six of the 204 newborn umbilical cord and cord blood samples are part of the 85 samples collected previously in *Dimas et al. (2009)*. The remaining 148 samples have been newly collected, and primary fibroblasts, EBV-immortalized lymphoblastoid cell lines and primary T-cells have been prepared and grown as stated previously (*Dimas et al., 2009*), with a few exceptions: the preparation of LCLs was done with 1 ml of 20 million re-suspended cells and 1 ml of EBV that were transferred to a 24-well plate. In addition, to confirm that the cord and cord blood acquired from the hospital belonged to the same individual, DNA was extracted from cord tissue and LCLs separately with the

Puregene cell kit (Gentra-Qiagen, Venlo, The Netherlands), microsatellite markers were assayed with QF-PCR and compared between the two samples coming from the same individual.

## Genotyping and imputation

We extracted DNA from LCLs with the QIAamp DNA Blood Mini Kit (Qiagen, Venlo, The Netherlands) and genotyped the 204 individuals with the Illumina 2.5M Omni chip. We applied several filters before imputation with PLINK v1.07 (*Purcell et al., 2007*). We filtered out any SNPs with >5% missing geno-types, minor allele frequency (MAF) <1%, gene call score <0.25 for more than 1% of samples, or Hardy-Weinberg equilibrium $p < 1 \times 10^{-6}$. This left 1,535,724 SNPs before imputation. We performed the imputation with Beagle v3.3.2 (*Browning and Browning, 2009*), into 21 million European panel SNPs of the 1000 genomes March 2012 release (*Abecasis et al., 2012*). From the final output, we required an $R^2 \geq 0.9$. This yielded a total of 6.9 million SNPs. For association analyses (eQTLs and mQTLs) we filtered out genetic outliers based on multidimensional scaling using HapMap populations (*Altshuler et al., 2010*; *Figure 1—figure supplement 1*), and we further filtered out any SNPs with MAF <5%, yielding sets of 5,209,348–5,278,330 SNPs.

## RNA sequencing and quantification

RNA was extracted from LCLs, fibroblasts and T-cells with RNeasy columns (Qiagen, Venlo, The Netherlands) or Trizol (Invitrogen, Carlsbad, CA). RNA samples were quantified with NanoDrop (Thermo Scientific, Waltham, MA) and Qubit (Invitrogen, Carlsbad, CA), and analyzed with a 2100 Bioanalyzer (Agilent, Santa Clara, CA). Samples were prepared for sequencing with the Illumina mRNA-seq and TruSeq sample preparation kits (Illumina, San Diego, CA) as indicated by manufacturer's instructions. In these library preparation procedures, poly-A RNA is selected using poly-T oligo-attached magnetic beads; then the RNA is cleaved, converted to first strand cDNA and after RNA digestion and second DNA strand synthesis, the fragments are end repaired and ligated to the adapters containing specific primer indexes. The cDNA libraries are then PCR amplified. Libraries were sequenced in either sets of 6 samples per lane in the Genome Analyzer II machine or 12 samples per lane in the HiSeq2000 machine, randomly pooling in the same lane samples from different cell-types and individuals. Afterwards, the 49-bp sequenced paired-end reads were mapped to the reference genome (*Lander et al., 2001*) GRCh37 with BWA v0.5.9 (*Li and Durbin, 2009*). Using SAMtools (*Li et al., 2009*), we kept reads mapping uniquely to the genome, with MAPQ ≥ 10 and properly paired. In order to quantify exons in a non-redundant way, we created a set of merged exons from the GENCODE v10 annotation (*Harrow et al., 2006*). In more detail, we took all protein coding and lincRNA transcripts and merged any overlapping exons into new exon units. We then counted the number of reads mapping to each exon unit (*Figure 1—figure supplement 2*). Technical outliers having less than 5M exonic reads or extremely low insert size mode were removed from the study. In order to keep only the set of exons that we considered expressed in each cell-type, we filtered out exons that had no reads mapped in 10% or more of the samples per cell-type. This yielded sets of 70,800–76,870 exons belonging to 12,265–12,863 expressed genes depending on the cell-type and analysis. Afterwards, we normalized the raw exon counts by scaling all libraries to 10 million reads, based on the total number of exonic reads per sample. In order to remove technical variance from our samples, we tested the effect of many different covariates on our scaled exon counts in each cell-type by linear regression. We decided to correct for mean GC content per library, run date, primer index and insert size mode, using a random effects model (*Figure 1—figure supplement 3*). Thus, our final exon expression levels are the residuals after correcting for those covariates plus the corrected mean (*Figure 1—figure supplement 4*).

## DNA methylation assay

DNA was extracted with the QIAmp DNA blood mini kit (Qiagen, Venlo, The Netherlands) and bisulfite converted (BC) using the Zymo Research EZ DNA MethylationTM Kit (using the 50-column or 96-well formats; Zymo Research, Irvine, CA). The starting quantity of DNA was 1000 ng and after BC we normalized all our samples to have a final concentration of 40 ng/µl or the closest possible amount. BC-DNA was then processed through the 450K Illumina Infinium HD Methylation Assay (Illumina, San Diego, CA) according to manufacturer's instructions. In brief, this is an assay that involves whole genome amplification, specific hybridization capture, and allele-specific single-base primer extension. Samples from the three cell-types were processed in a randomized manner. The controls available in the methylation assay for verifying the efficiency of our bisulfite conversion indicated that none of our samples had bad efficiency of bisulfite conversion. We removed three technical outliers that didn't cluster well with any cell-type in a principal component analysis. From the 482,421 CpG sites assayed, we filtered out

probes with 1000 genomes SNPs or indels at European minor allele frequency >5% and then any other probes that had any of the final set of SNPs used in our study. This yielded a total of 416,118 CpG sites to work with. We quantile normalized the data in the following way. The Illumina 450k array is composed of two probe types. Probe type 1 reads the methylated signal and unmethylated signal in the same channel (red or green) and probe type 2 reads the methylated signal in the green channel and the unmethylated signal in the red channel. The distribution of β-values before any normalization is quite different for each probe type, thus the two probe types were normalized separately. For probe type 1, the intensity for the methylated and unmethylated signals were quantile normalized across individuals for each color channel separately. As the density of signal intensities showed that there was a color bias (at least for the unmethylated probe), the color channels were quantile normalized within individuals for the methylated signal and unmethylated signal separately. For probe type 2, the methylated signals and unmethylated signals were quantile normalized across individual. As for this probe the methylated signal is always in green and the unmethylated signal is always in red. The color bias needs to be corrected before computing β-values. In order to correct the color bias, the methylated (unmethylated) signal probe type 2 normalized across individuals is quantile normalized with the methylated (unmethylated) signal of probe type 1 that was previously quantile normalized across individuals and within for color correction. In a final step, the β-values of the two probe types were quantile normalized together leading to the final β-values (*Figure 1—figure supplements 5–7*). The β-value is defined as the proportion of fluorescent signal from the methylated allele over the total fluorescent signal (methylated and unmethylated alleles); this represents the percentage of methylation per site (*Bibikova et al., 2006*).

## Association analyses and multiple testing correction

We performed Spearman rank correlations (SRC) between (x and y variables): SNP genotypes and exon expression levels (eQTLs) in 183–185 samples using a 1-MB window to either side of the TSS, SNP genotypes and CpG site methylation levels (mQTLs) in 66–111 samples using a 5-kb window to either side of the CpG site, CpG site methylation levels and exon expression levels (eQTMs) in 66–118 samples using a 50-kb window to either side of the TSS. Different window sizes were tested for mQTL and eQTM analyses using a subset of 56 individuals. The final window size chosen is the one that yielded the highest enrichment of significant associations. As previously mentioned, association analyses involving genetic variation were ran with SNPs with MAF ≥ 5% and excluding genetic outlier individuals (*Figure 1—figure supplement 1*). Similar to previous analyses (*Stranger et al., 2007*; *Montgomery et al., 2010*), for each type of association analysis and each cell-type the phenotype belonging to the y variable was permuted 1000 times and the median p value distribution was summarized among the genes or CpG sites. For expression levels this was done on all exons belonging to 1000 randomly selected genes and for each gene the minimum p value distribution out of all its exons was chosen. For methylation levels, 50,000 random methylation probes were selected. Based on the null p value distributions yielded by the permutations, we selected for each analysis the associations that passed the permutation threshold that yielded 10% or less False Discovery Rate (FDR) in a per gene/CpG basis.

## DNA methylation differentiation metric

We have used the coefficient of variation (CV) as a measure of DNA methylation differentiation level, as it is a variability measure that controls for the mean methylation. In detail, for each site we have calculated the standard deviation of the median methylation in each cell-type, divided by the mean of the three medians. By using the median methylation level per site for each cell-type, our differentiation measure is estimating the between cell-type variation only, not taking into account the within cell-type variation.

## Context specific analyses

Enhancers and CTCF peaks, were downloaded from the UCSC genome browser tables (*Browning and Browning, 2009*) and come from ChIP-seq experiments of the ENCODE project (*Birney et al., 2007*; *Rosenbloom et al., 2010*; *Bernstein et al., 2012*) and specific groups (*Boyle et al., 2008*; *Ernst and Kellis, 2010*; *Ernst et al., 2011*; *Thurman et al., 2012*). For LCLs we used data belonging to the cell line GM12878. For fibroblasts we used data belonging to the cell line NHLF. Unfortunately, there was limited data available for T-cells so we used the GM12878 data, given its close lineage relatedness. CpG islands (CGIs) were downloaded from the UCSC genome browser (*Karolchik et al., 2004*). The promoter region is defined as going from -1kb of the TSS to +2kb of the TSS, based on methylation

pattern observed in this region (*Figure 1—figure supplement 7A*). CGI promoters are defined as promoter regions overlapping any CGI.

The genomic feature enrichment analyses of eQTLs and mQTLs were done by comparing the number of observed overlaps with those found in a null set. The null set selects SNP-exon or SNP-CpG pairs with MAF, distance and expression or methylation values matching the distribution of eQTLs and mQTLs, respectively, from genes or CpG sites lacking any significant associations. The differential genomic feature overlap analyses of eQTMs were done by comparing the number of observed overlaps for positive and negative eQTMs, with the number of expected overlaps in non-eQTM sites. Fisher's exact tests were used for assessing the significance of each analysis.

Intersections of genomic features were done with BEDTools v2.7.1 (*Quinlan and Hall, 2010*).

## Allele specific expression analysis

The allelic imbalance (i.e., absolute distance from the expected 0.5 ratio) was measured for assayable heterozygote sites. First, we excluded sites that are susceptible to allelic mapping bias: 1) sites with 50-bp mapability (*Karolchik et al., 2004*) <1 implying that the 50-bp flanking region of the site is non-unique in the genome, and 2) simulated RNA-Seq reads overlapping the site show >5% difference in the mapping of reads that carry the reference or non-reference allele. In all the analyses, we used mapping and base quality threshold ≥10. Next, we calculated the expected reference allele ratio for each individual by summing up reads across all sites separately for each SNP allele combination after down-sampling reads of sites in the top 25th coverage percentile in order to avoid the highest covered sites having a disproportionally large effect on the ratios. These expected REF/TOTAL ratios (typically 0.494–0.517 as measured by 10th and 90th quantiles across samples) correct for remaining slight genome-wide mapping bias as well as GC bias in each individual. Finally, for all the sites covered by ≥16 reads in each individual, we calculated the absolute distance from the expected 0.5 ratio described above. Because our genotypes included imputed data, we verified the genotype from the RNA sequencing data by requiring the observation of both of the SNP alleles in at least one cell type per individual.

The number of genes in the homozygote and heterozygote eQTL analysis in *Figure 3A* are 1473 and 1453 in fibroblasts, 2197 and 2177 in LCLs, and 1288 and 1284 in T-cells, respectively. The number of genes in the homo-methylated and semi-methylated eQTM analysis in *Figure 3A* are 279 and 106 in fibroblasts, 1738 and 794 in LCLs, 1544 and 969 in T-cells, respectively. The number of genes in the imprinted genes analysis in *Figure 3A* are 29, 17, 15 in fibroblasts, LCLs and T-cells, respectively. For measuring allelic imbalance in imprinted genes, all the filtering and requirements mentioned above were applied except we did not require that both SNP alleles are seen. Imprinted genes were taken from http://www.geneimprint.com.

## Interaction permutation analysis

The expression values of the exon-SNP-methyl triplets tested were permuted 1000 times to assess the expected p value distributions for the interaction between genetic variation and DNA methylation on gene expression. For each round of permutations a $\pi_1$ was calculated, creating a null distribution of $\pi_1$ from which an empirical p value can be inferred (as reported in the main text). Additionally, the 95% confidence intervals were calculated from the 1000 permutations of all tests in each cell-type. False discovery rates of the reported significant results were calculated using the QVALUE R package (*Storey and Tibshirani, 2003*).

## Causative model analysis

Bayesian networks (BN) are directed acyclic graphs where nodes represent random variables and edges represent the conditional dependences among nodes. The direction of the edges between two nodes can be interpreted as a causal relationship. BN have been used before to infer causality in genetic problems (*Schadt et al., 2005*; *Zhu et al., 2008*). Likelihood methods are commonly used to estimate the most likely network—that is, the set of causal relationships among the different variables that better agrees with the data.

In a BN every node has an associated probability function and, together with the conditional dependencies represented by the edges, they conform the joint probability density of the network. BN satisfy the local Markov property—that is, each variable is conditionally independent of its non-descendants given its parent variables. The Markov property defines the decomposition of the joint probability density of the network into a set of local distributions. Thanks to the Markov property it is

easy to calculate the likelihood of a given BN. Since different networks have different complexities, it is common to use a score that takes into account the network complexity instead of the raw likelihood to compare the networks. We used the R package bnlearn (*Scutari, 2010*) to calculate the maximum likelihood of the networks and the Akaike Information Criterion (AIC) score. AIC = 2k − 2ln(L), where k is the number of parameters and L is the maximum likelihood. To calculate how much better one network is compared to another, we used the relative likelihood of one network with respect to the other. If we have two networks, N1 and N2, with fixed parameters and AIC(N1) ≤ AIC(N2), then the relative likelihood of N2 with respect to N1 is defined as: exp((AIC(N1)−AIC(N2))/2). The networks we have compared (mentioned in main text) have five parameters each.

To test the networks, we selected three datasets that were then merged and normalized for size in order to calculate the relative frequencies of the inferred best network. In the first dataset we selected the best eQTM and for that methylation site the best mQTL, not requiring statistical significance for the eQTL (n = 219, 1074 and 1128 in fibroblasts [F], LCLs [L], and T-cells [T], respectively), which would enrich for the SME model since the eQTL is the most distant association. In the second dataset, for each exon we selected its best eQTL and its best eQTM; not requiring significance for the mQTL (n = 287, 1316 and 871 in F, L, T), which would enrich for the SEM model since the mQTL is the most distant association. And in the third, for each exon we selected its best eQTL and kept the cases where the SNP was the best mQTL for a methylation site; not requiring significance for the eQTM, which would enrich for the INDEP model since the eQTM is the most distant association (n = 413, 726, 775, in F, L, T).

In order to create a list of high confidence calls, we have made use of a semi-parametric method, the Causal Inference Test (*Millstein et al., 2009*), that requires a level of significance for calling two out of the three models. The CIT makes SME and SEM calls if either has a p<0.05, makes no call if p<0.05 for both models (highly infrequent) and makes a call for the INDEP if neither SME or SEM has a significant p-value. We have decided to call a high confidence set of SME and SEM models when they are called as such by both the BN and the CIT methods. Since the INDEP calls in the CIT do not require significance, we called a high confidence INDEP if 1) it is called INDEP by both methods, and 2) its BN relative likelihood (RL) to the second best model is equal or higher to the median RL of the SME and SEM high confidence calls to their second best prediction.

## Transcription factor expression to methylation level analyses

As explained in the text, the transcription factor (TF) expression levels were Spearman rank correlated to methylation levels of eQTM sites falling in their reported TF binding sites (*Bernstein et al., 2012*; *Gerstein et al., 2012*), excluding any sites within 1 Mb of the TF gene transcription start site. The number of expressed TFs in each cell-type is 94, 106 and 105, comprising 666, 851 and 776 exons in fibroblasts, LCLs and T-cells, respectively. The number of targeted eQTM methylation sites in each cell-type is 459, 3832 and 3497 in fibroblasts, LCLs and T-cells, respectively. The number of tests performed is 17,747, 209136 and 152263 in fibroblasts, LCLs and T-cells, respectively.

To show the expected uniform distribution of p values for these correlations and compare it with the (highly enriched) observed, we have permuted once the TF expression values for each TF-meth correlation in each cell-type and plotted the p-value distributions (*Figure 4—figure supplement 2*). To call significant cases for the target sites of each TF, the false discovery rate was calculated for each using the QVALUE R package (*Storey and Tibshirani, 2003*).

We tested the interactions between SNPs and TF abundance on their effect on DNA methylation by taking the top mQTL SNPs that fell in TF peaks and whose associated methylation site correlated significantly with the TF expression level of that peak (10% FDR for both types of associations). We excluded cases in which the SNP and the TF abundance were correlated (p<0.05) and in which the number minor allele homozygotes was less than four. This yielded a total set of 114 SNP-TF-meth triplets for which the interaction between SNP and TF abundance was tested by linear regression (*Figure 4—figure supplement 1*).

## Data availability

Genotyping, RNA-seq and DNA methylation data have been submitted to the EMBL-EBI European Genome-Phenome Archive (https://www.ebi.ac.uk/ega/) under accession number EGAS00001000446. Zipped text file tables with the identified eQTLs, mQTLs and eQTMs for each cell-type are available as *Table 1—source data 1*. Zipped text file tables with the high confidence SME, SEM and INDEP calls for each cell-type are available as *Figure 4—source data 1*.

## Acknowledgements

We thank the genomics platform of the University of Geneva, VITAL-IT and the data producers from the ENCODE Consortium (*Bernstein et al., 2012*). We would also like to thank Helena Kilpinen, Alexandra C Nica, Daniel Robyr, Reza Sailani and Federico Santoni for comments and useful discussions.

## Additional information

### Competing interests

SEA: Senior editor, *eLife*. ETD: Reviewing editor, *eLife*. The other authors declare that no competing interests exist.

### Funding

| Funder | Grant reference number | Author |
|---|---|---|
| European Research Council (ERC) | 249968 | Stylianos E Antonarakis |
| European Research Council (ERC) | 260927 | Emmanouil T Dermitzakis |
| Swiss National Science Foundation | 144082 | Stylianos E Antonarakis |
| Swiss National Science Foundation | 31003A_130342, CRSI33_130326 | Emmanouil T Dermitzakis |
| Louis-Jeantet Foundation | | Emmanouil T Dermitzakis |
| Blueprint | European Commission-FP7 | Stylianos E Antonarakis, Emmanouil T Dermitzakis |

The funders had no role in study design, data collection and interpretation, or the decision to submit the work for publication.

### Author contributions

MG-A, Acquisition of data, Analysis and interpretation of data, Drafting or revising the article; TL, AB, HO, AY, JB, Analysis and interpretation of data, Drafting or revising the article; SBM, SEA, ETD, Conception and design, Analysis and interpretation of data, Drafting or revising the article; TG, AL, PM, Analysis and interpretation of data; LR, AP, EF, DB, MG, IP, CB, MG, CG, Acquisition of data

### Ethics

Human subjects: Informed consent was obtained from all subjects. The local ethics committee at University Hospitals of Geneva has approved this project (CER 10-046).

## Additional files

### Major datasets

The following dataset was generated:

| Author(s) | Year | Dataset title | Dataset ID and/or URL | Database, license, and accessibility information |
|---|---|---|---|---|
| Dermitzakis Emmanouil T et al. | 2013 | Gencord | EGAS00001000446; https://www.ebi.ac.uk/ega/studies/EGAS00001000446 | Publicly available at European Genotype Phenotype Archive. |

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
