## [Decision Letter]

Thank you for choosing to send your work entitled “Passive and active DNA methylation and the interplay with genetic variation in gene regulation” for consideration at *eLife*. Your article has been evaluated by a Senior editor and 2 reviewers, one of whom is a member of our Board of Reviewing Editors.

The Reviewing editor and the other reviewer discussed their comments before we reached this decision, and the Reviewing editor has assembled the following comments to help you prepare a revised submission.

The study tackles an important and timely set of questions in the nexus of DNA methylation, gene regulation, and genetic variation. The genomics dataset collected in this study is exemplary and, indeed, is one of the first of the next generation of “genetics–genomics” studies, with three cell types (two of them primary) isolated from over 200 newborns, all genotyped, and profiled for expression (RNA-Seq) and DNA methylation.

In this sense, all of us would be delighted to see the data published, and with the appropriate analytics and conclusions this could be an important paper, well within the scope of *eLife*. Indeed, it is precisely because the dataset is likely to be broadly used, because the design could be deployed in many other contexts by other groups, and because the conclusions address fundamental questions in gene regulation, that we feel that it is critical that the analytics underlying the conclusion be very carefully and very strongly supported.

In this respect, unfortunately, we found the current manuscript lacking in two important ways:

1) Very strong and broad conclusions were drawn as we detail below, but they were not equally strongly supported in the actual analyses/data. These should be toned down to the appropriate level, with some probably removed or replaced with hypothesis level assertions; and

2) In multiple places throughout the work, the statistics need to be performed more carefully, with appropriate controls, which we detail specifically below.

3) Next, once the statistical analyses are done appropriately, the authors are likely to remain with a far fewer number of significant loci/genes, but those would be individually significant, and the authors should highlight some of those specific examples, instead of focusing solely on aggregate numbers as currently written.

4) Finally, since some of the study’s inherent value is as a resource, it is critical that the manuscript provides processed results in summary tables (as source data files), listing all genes, associated SNPs, and other traits. It is also critical to state clearly within the manuscript where and how the data can be obtained by others (with the expected constraints associated with human genotype data).

We further detail below how to address points 1 and 2, in the context of each of the seven major claims of the paper. We emphasize that without a serious analytic revision, the manuscript will not be appropriate for publication in *eLife*.

We identify seven major claims in the paper. We list each here and its extent of support, and make specific suggestions for how to address each analytic shortcoming.

* Claim 1 addresses eQTMs, the relation between variation in methylation and expression. The authors draw the conclusion that eQTMs are mostly independent of the mechanism involved in the repressive effect of DNA methylation on expression across genes. This is a strong conclusion, but is currently poorly supported. Specifically, the eQTMs were partitioned into only two bins (positive and negative effect). It might be, however, that most of the pos- and neg-eQTM effects are insignificant (around 0) and therefore some important inter-relations are masked within a sea of noise. Instead, the same plots should be regenerated for eQTMs split into a larger number of bins (e.g., 5 bins: very-neg, neg, no-sign, pos, very-pos-eQTMs), to show whether the negative correlation holds in all of these 5 bins, and more generally is not an artifact of the binning.

* Claim 2 addresses differentially methylated regions (DMRs) across cell types and asked whether DMRs are more often associated to eQTMs and mQTLs than non-DMRs.

First, the relation between DMRs and eQTMs might be a byproduct of their relation with the average methylation. This is because the higher the average, the higher the variance, and thus we may get a higher differentiation per methylation. To control for this possibility the authors should use a 'normalized differentiation in methylation' (i.e., use the residuals of differentiation per methylation after removing the effect of the methylation average), and then use these normalized (residual) values to repeat the same test relating (residual) differentiation with mQTLs and eQTMs.

Second, regardless of the validity of the test, the additional biological conclusions are over-stretched and should be toned down.

* Claim 3 associates mQTLs and eQTMs with different features. The differential association of mQTLs to CGI and non-CGI promoters is intriguing and well supported. The analysis of pos- and neg-eQTMs should be repeated with finer binning (as claim 1, above), but is otherwise satisfactory.

* Claim 4 shows lack of support for a contribution of methylation to allele specific expression, independent of sequence variation. This is a strong assertion, but is also a negative result. Our main concern is that it is hard to devise an appropriate control here. We urge the authors to remove this result, unless they can provide additional support.

* Claim 5 tests for synergistic interactions between genetic variants and DNA methylation on gene expression. Although the authors note that they “controlled” for artificial inflation, Figure 3 shows that there is a clear inflation in the resulting *P*-values. What is the reason for this inflation? Without correcting for this, it is impossible to judge the number of significant relations. We advise that the authors use at least bootstrapping for some empirical correction, and preferably correct for population structure and other confounders.

* Claim 6 concerns different causality models and their proportions in the different cell types. Each of the three models has a different number of parameters, and the correction with AIC is based on theoretical considerations and hence might not compensate for the different degrees of freedom. In addition, there is no evaluation of the significance of the resulting predicted models (which then precludes the authors – and future readers – from following up on individual findings). To correct for such potential artifacts, we suggest applying a direct evaluation of the distribution of relative likelihood for each pair of models based on data reshuffling (permutation test). Based on such tests, it is possible to (A) get a threshold for each of the likelihood comparisons using a common *P*-value threshold. This would allow comparing results from different cell types and models using a common systematic threshold; and (B) discriminate significant vs. non-significant relative likelihood scores. Once this is done the authors should report how many of the results are significant and what the statistics are when using significant predictions only, and, finally, they should highlight specific examples, which would enhance the paper's impact and biological relevance.

* Claim 7 tries to couch the findings in a mechanistic setting, claiming that TF levels modulate the effect of DNA sequence variation. This is an intriguing and important direction, but it has to be done carefully.

First, as above, it is not clear whether the reported *P*-values are corrected for the special characteristics of the data. Applying a permutation test and deriving a corrected *P*-value based on such an analysis would allow evaluating the statistical significance of the reported results. Furthermore, while we do not require the authors to conduct follow up biological experiments, in the absence of such follow up, the conclusions should be toned down significantly.

---

## [Author Response]

1) Very strong and broad conclusions were drawn as we detail below, but they were not equally strongly supported in the actual analyses/data. These should be toned down to the appropriate level, with some probably removed or replaced with hypothesis level assertions; and

We have toned down the conclusions and performed additional analyses to further support some of our results, as detailed below.

*2) In multiple places throughout the work, the statistics need to be performed more carefully, with appropriate controls, which we detail specifically below*.

We have dealt with the concerns below either by improving the statistics or by clarifying why the statistical analyses used in the original submission were appropriate and further clarified in the text when necessary.

*3) Next, once the statistical analyses are done appropriately, the authors are likely to remain with a far fewer number of significant loci/genes, but those would be individually significant, and the authors should highlight some of those specific examples, instead of focusing solely on aggregate numbers as currently written*.

We have incorporated in the main text specific examples, as detailed in the response to claim 6.

*4) Finally, since some of the study’s inherent value is as a resource, it is critical that the manuscript provides processed results in summary tables (as source data files), listing all genes, associated SNPs, and other traits. It is also critical to state clearly within the manuscript where and how the data can be obtained by others (with the expected constraints associated with human genotype data)*.

We have created tables as text files for the discovered eQTLs, mQTLs and eQTMs in each cell-type as a single zipped file as [Supplement-material SD1-data]. We have also generated a zipped file as [Supplement-material SD2-data] in which we report in text tables the high confidence calls of the inferred mechanistic relationships as detailed in claim 6.

The transcriptome, genotyping, and DNA methylation array data will be available in the EMBL-EBI European Genome-Phenome Archive under accession number EGAS00001000446.

We have described the location details of the primary data and supplementary files in Materials and methods under “Data availability”.

*We identify seven major claims in the paper. We list each here and its extent of support, and make specific suggestions for how to address each analytic shortcoming*.

** Claim 1 addresses eQTMs, the relation between variation in methylation and expression. The authors draw the conclusion that eQTMs are mostly independent of the mechanism involved in the repressive effect of DNA methylation on expression across genes. This is a strong conclusion, but is currently poorly supported. Specifically, the eQTMs were partitioned into only two bins (positive and negative effect). It might be, however, that most of the pos- and neg-eQTM effects are insignificant (around 0) and therefore some important inter-relations are masked within a sea of noise. Instead, the same plots should be regenerated for eQTMs split into a larger number of bins (e.g., 5 bins: very-neg, neg, no-sign, pos, very-pos-eQTMs), to show whether the negative correlation holds in all of these 5 bins, and more generally is not an artifact of the binning*.

We believe that there has been some misunderstanding since the assessment of positive and negative eQTMs is only done for statistically significant eQTMs. Specifically, due to our multiple testing corrections, in order to call an eQTM, we require a high level of nominal significance (as indicated in Table 1) controlling for multiple testing with an FDR of 10%. As a consequence, the strength of the correlations of eQTMs as measured by the Spearman correlation coefficient are far from zero (see histograms below: Figure 5), the minimum of the absolute value being 0.368, 0.308, and 0.407 in fibroblasts, LCLs, and T-cells, respectively. Furthermore, for the effect size of eQTMs as the slope of the linear regression of expression explained by DNA methylation, we also obtain values far from zero. The minimum of the absolute effect size is 3.16 and 7 in fibroblasts and T-cells, and LCLs have only 6 eQTMs with effect size below 1. Hence, our eQTMs do not contain insignificant associations, and thus the further binning is not necessary.

Moreover, although we are dealing with 1140 pos-eQTMs and 5112 neg-eQTMs at promoter regions, these involve only a limited number of genes (703 and 1758 for pos and neg-eQTMs, respectively). Hence, binning our eQTMs would lead to a smaller number of genes per bin and consequently a lower dynamic range of gene expression levels, which would not allow a proper assessment of across gene expression.

Finally, we would like to highlight that the reported strength of the negative correlation of pos and neg-eQTM median methylation with across gene expression (rho = -0.11 and -0.10 for pos and neg-eQTMs, respectively) is comparable to the strength of the correlations between promoter DNA methylation and across gene expression measured per individual for all expressed genes, as illustrated in Figure 2—figure supplement 1. Hence, the correlation we report in Figure 2 is as robust as that found at a genome-wide level.

Specific points in the paper have been clarified to reflect these issues.

** Claim 2 addresses differentially methylated regions (DMRs) across cell types and asked whether DMRs are more often associated to eQTMs and mQTLs than non-DMRs*.

*First, the relation between DMRs and eQTMs might be a byproduct of their relation with the average methylation. This is because the higher the average, the higher the variance, and thus we may get a higher differentiation per methylation. To control for this possibility the authors should use a ‘normalized differentiation in methylation’ (i.e., use the residuals of differentiation per methylation after removing the effect of the methylation average), and then use these normalized (residual) values to repeat the same test relating (residual) differentiation with mQTLs and eQTMs*.

*Second, regardless of the validity of the test, the additional biological conclusions are over-stretched and should be toned down*.

The measure of DNA methylation differentiation that we are using (coefficient of variation, CV) already controls for the bias that the reviewers point out, that of a higher mean having a higher variance. We calculate the CV by measuring the standard deviation among the three medians of the methylation level per site in each cell-type divided by the mean of these three medians. In this manner, our differentiation measure is not positively correlated with average methylation as shown in the plot below (in fact, it is negatively correlated: Figure 6).

We would also like to emphasize that by using the median methylation level per site for each cell-type, our differentiation measure is estimating the between cell-type variation only, not taking into account the within cell-type variation. We have clarified these points in the main text, and in Materials and methods by including a section called “DNA methylation differentiation metric”. Additionally, we have toned down the conclusion of this analysis in the main text.

** Claim 3 associates mQTLs and eQTMs with different features. The differential association of mQTLs to CGI and non-CGI promoters is intriguing and well supported. The analysis of pos- and neg-eQTMs should be repeated with finer binning (as claim 1, above), but is otherwise satisfactory*.

As explained above that the eQTMs are statistically significant and of substantial effect size, binning is not meaningful in this analysis.

** Claim 4 shows lack of support for a contribution of methylation to allele specific expression, independent of sequence variation. This is a strong assertion, but is also a negative result. Our main concern is that it is hard to devise an appropriate control here. We urge the authors to remove this result, unless they can provide additional support*.

We understand the reviewers’ concern and have re-arranged the figure and added an additional positive control in order to better support our results (Figure 3). We are of the opinion that this result is important and while not definitive it is still worth presenting with the necessary caveats.

As pointed out by the reviewers, our results show a lack of support for a contribution of methylation to allele specific expression, independent of sequence variation when comparing homomethylated eQTMs versus semimethylated eQTMs. This is contrary to the expected genetic effect, which leads to a significantly higher allelic imbalance in heterozygous compared to homozygous eQTLs. While the results as presented previously appeared as a “negative” result, we would like to point out that the allelic imbalance of heterozygous eQTLs is significantly greater than the allelic imbalance of semimethylated eQTMs (*P* = 7.50E-03, *P* = 3.40E-07, *P* = 1.1E-13 in fibroblasts, LCLs and T-cells, respectively), hence it is a positive result. Furthermore, we have incorporated a positive control of methylation effects on allelic imbalance, by measuring the allelic imbalance in genes reported to be imprinted, which represent a direct measure of ASE caused by an allelic signal of DNA methylation. As seen in the white boxplots of Figure 3, allelic imbalance in imprinted genes is significantly higher than in homomethylated eQTMs (with *P* = 6.93E-34, *P* = 9.66E-04, *P* = 8.48E-05, in fibroblasts, LCLs and T-cells, respectively).

We have integrated these results into the main text, Figure 3, and the Materials and methods. Moreover, we have highlighted in the Discussion the need for further studies to confirm these findings. We hope the reviewers find this additional support satisfactory and we remain open to any further suggestions that may arise.

** Claim 5 tests for synergistic interactions between genetic variants and DNA methylation on gene expression. Although the authors note that they “controlled” for artificial inflation,*
Figure 3
*shows that there is a clear inflation in the resulting* P*-values. What is the reason for this inflation? Without correcting for this, it is impossible to judge the number of significant relations. We advise that the authors use at least bootstrapping for some empirical correction, and preferably correct for population structure and other confounders*.

We understand the reviewers’ concern on the observed inflation in the resulting *P*-values and thank them for pointing that out. We can discard population structure confounders since we have already corrected for it (in all our analyses involving genetic variation) by excluding the genetic outliers depicted in Figure 1—figure supplement 1, as determined by multidimensional scaling genetic clustering using several HapMap populations. However, although we eliminated from our analyses any SNP-methyl correlation with *P* < 0.05, there could still be low levels of correlation that could potentially still inflate our *P*-values.

In order to better control for this, we have permuted the expression values of all the triplets tested, 1000 times, keeping SNP and methyl pairs of values in their original form. This allows us to maintain the correlation structure between the SNP and the methylation site and any inflation due to that, while randomizing the expression. From these permutations, we have derived the 95% confidence intervals and incorporated them into the qqplots of Figure 3. The interaction *P*-values for fibroblasts fail to cross the upper 95^th^ confidence limit, while for T-cells and LCLs the observed *P*-values remain mostly or entirely above the upper 95^th^ confidence limit, respectively. Additionally, for each round of permutation of the tested triplets, we have calculated the π1 statistic. Hence, by creating a null distribution of π1 we can assess how different it is to the observed π1. The observed π1 of each cell-type was highly significant (all *P*-values < 0.001) with the top permuted π1 being 7.2, 8.5 and 2 times smaller than the observed π1 in fibroblasts, LCLs and T-cells, respectively.

We have included these results in the main text and Figure 3, and clarified the details in Materials and methods under “Interaction permutation analysis”.

** Claim 6 concerns different causality models and their proportions in the different cell types. Each of the three models has a different number of parameters, and the correction with AIC is based on theoretical considerations and hence might not compensate for the different degrees of freedom. In addition, there is no evaluation of the significance of the resulting predicted models (which then precludes the authors – and future readers – from following up on individual findings). To correct for such potential artifacts, we suggest applying a direct evaluation of the distribution of relative likelihood for each pair of models based on data reshuffling (permutation test). Based on such tests, it is possible to (A) get a threshold for each of the likelihood comparisons using a common* P*-value threshold. This would allow comparing results from different cell types and models using a common systematic threshold; and (B) discriminate significant vs. non-significant relative likelihood scores. Once this is done the authors should report how many of the results are significant and what the statistics are when using significant predictions only, and, finally, they should highlight specific examples, which would enhance the paper’s impact and biological relevance*.

We are not sure why the reviewers suggest that there is a different number of parameters for each model. The number of parameters of a discrete Bayesian network is defined as the sum of the number of logically independent parameters of each node given its parents (Chickering, 1995). We have 5 parameters in each of our three models, which we have now specified in the Materials and methods. Furthermore, in order to test an exon-SNP-methyl triplet we require that at least two of the three pair-wise correlations are significant; hence, permuting the data would break these prior requirements and would not truly assess the confidence of the called model. However, we understand the reviewers’ concern on better grasping the relative likelihood space between the models and reporting the number of significant predictions. In order to do this we have: 1) presented the general landscape of relative likelihoods of the models, 2) created a list of high confidence calls based on the overlap of two distinct methods.

For point (1), instead of calling the best model and then presenting the relative frequencies of each (previously Figure 4, now Figure 4—figure supplement 1), we present directly the relative likelihood space for the three models across all exon-SNP-methyl triplets tested. As seen in the new plot in Figure 4, the relative contribution of the three models remains very similar to the one obtained before, with a slight general increase of pos-eQTM likelihood.

For point (2), creating a list of high confidence calls, we have made use of a semi-parametric method, the Causal Inference Test (CIT; Millstein et al. 2009), that requires a level of significance for calling two out of the three models. The CIT makes SME and SEM calls if either has a *P* < 0.05, it makes no call if *P* < 0.05 for both models (highly infrequent), and it makes a call for the INDEP if neither SME or SEM has a significant *P*-value. We have decided to call a high confidence set of SME and SEM models when they are called as such by both the Bayesian Networks (BN) and the CIT methods. Since the INDEP calls in the CIT do not require significance, we called a high confidence INDEP if 1) it is called INDEP by both methods, and 2) its BN relative likelihood (RL) to the second best model is equal or higher to the median RL of the SME and SEM high confidence calls to their second best prediction.

From the total number of tests, 61%, 36%, and 27% were called as high confidence (HC) in fibroblasts, LCLs, and T-cells, respectively (see details in table below). The relative frequencies of these HC calls have been added as Figure 4—figure supplement 1 (they look similar to the general landscapes). We have reported these results in the main text and we have described the details in the Materials and methods. Additionally, we provide text file tables of the high confidence calls in a zipped file as [Supplement-material SD2-data], in which the exon-SNP-methyl triplets are reported, as well as the Spearman rho value for inferring the sign of the correlation between expression and DNA methylation, and a column indicating the RL of the inferred model over the second best BN prediction, so that users can further filter by confidence if needed.Cell-typeNum TestsHC SMEMedian RL to 2ndHC SEMMedian RL to 2ndHC INDEPINDEP RL threshold**L**29281934.51396.17175.1**F**831442.8364.64283.5**T**25842925.1673.13455.0

Finally, we have described in the main text specific examples of a high confidence call for each of the three models.

** Claim 7 tries to couch the findings in a mechanistic setting, claiming that TF levels modulate the effect of DNA sequence variation. This is an intriguing and important direction, but it has to be done carefully*.

*First, as above, it is not clear whether the reported P-values are corrected for the special characteristics of the data. Applying a permutation test and deriving a corrected P-value based on such an analysis would allow evaluating the statistical significance of the reported results. Furthermore, while we do not require the authors to conduct follow up biological experiments, in the absence of such follow up, the conclusions should be toned down significantly*.

The correlations between TF abundance and DNA methylation at their binding sites are calculated with Spearman rank correlation and therefore the P-values are expected to behave appropriately. However, for our and the reviewers’ peace of mind, we have permuted the TF expression values for each TF-methyl correlation in each cell-type. We now present the expected null P-value distribution (under permutation), which as expected was uniform, and the one we observe (with their associated π1 statistics) in the histograms added to Figure 4—figure supplement 2. We have also specified in the Materials and methods the usage of Spearman rank for these correlations and included a description of the permutation procedure.

Finally, we have also toned down the conclusions in the text.